# Development and Clinical Evaluation of Serum and Urine-Based Lateral Flow Tests for Diagnosis of Human Visceral Leishmaniasis

**DOI:** 10.3390/microorganisms9071369

**Published:** 2021-06-23

**Authors:** Sarfaraz Ahmad Ejazi, Somsubhra Thakur Choudhury, Anirban Bhattacharyya, Mohd Kamran, Krishna Pandey, Vidya Nand Ravi Das, Pradeep Das, Fernando Oliveira da Silva, Dorcas Lamounier Costa, Carlos Henrique Nery Costa, Mehebubar Rahaman, Rama Prosad Goswami, Nahid Ali

**Affiliations:** 1CSIR-Indian Institute of Chemical Biology, Kolkata 700032, India; sarfaraz.ejazi@hotmail.com (S.A.E.); sthakurchoudhury@gmail.com (S.T.C.); anirbanbiomedical@gmail.com (A.B.); mohdkamran9808@gmail.com (M.K.); 2Rajendra Memorial Research Institute of Medical Sciences, Patna 800007, India; drkrishnapandey@yahoo.com (K.P.); dasvnr@icmr.org.in (V.N.R.D.); drpradeep.das@gmail.com (P.D.); 3Department of Community Medicine, Universidade Federal do Piaui, Teresina 64001-450, Brazil; laboratorioprovidapi@gmail.com (F.O.d.S.); dorcas.lc@gmail.com (D.L.C.); chncosta@gmail.com (C.H.N.C.); 4School of Tropical Medicine, Kolkata 700073, India; rmehbub@gmail.com (M.R.); drrpgoswami@gmail.com (R.P.G.)

**Keywords:** *Leishmania*, diagnosis, lateral flow assay, serology, urine

## Abstract

Visceral leishmaniasis (VL), a fatal parasitic infection, is categorized as being neglected among tropical diseases. The use of conventional tissue aspiration for diagnosis is not possible in every setting. The immunochromatography-based lateral flow assay (LFA) has attracted attention for a long time due to its ability to give results within a few minutes, mainly in resource-poor settings. In the present study, we optimized and developed the LFA to detect anti-*Leishmania* antibodies for VL diagnosis. The performance of the developed test was evaluated with serum and urine samples of Indian VL patients and Brazilian sera. The new test exploits well-studied and highly-sensitive purified antigens, LAg isolated from *Leishmania donovani* promastigotes and protein G conjugated colloidal-gold as a signal reporter. The intensity of the bands depicting the antigen–antibody complex was optimized under different experimental conditions and quantitatively analyzed by the ImageJ software. For the diagnosis of human VL in India, LFA was found to be 96.49% sensitive and 95% specific with serum, and 95.12% sensitive and 96.36% specific with urine samples, respectively. The sensitivity and specificity of LFA were 88.57% and 94.73%, respectively, for the diagnosis of Brazilian VL using patients’ sera infected with *Leishmania infantum*. LFA is rapid and simple to apply, suitable for field usage where results can be interpreted visually and particularly sensitive and specific in the diagnosis of human VL. Serum and urine LFA may improve diagnostic outcomes and could be an alternative for VL diagnosis in settings where tissue aspiration is difficult to perform.

## 1. Introduction

Visceral leishmaniasis (VL) or kala-azar is a serious yet neglected parasitic disease of tropical and sub-tropical regions [1]. Historically, the disease has been endemic mainly in the Indian Subcontinent, Latin America and East Africa [2]. However, the global extent of this disease is continuously changing, with reports of the disease arising from the newer areas of Southern Europe and Mediterranean regions [3]. Successfully employing VL control measures early and accurate, field-adaptable diagnoses are imperative, along with safe and efficacious treatment and vector control strategies [4]. Early symptoms arousing suspicion of VL can overlap with other febrile illnesses such as malaria and dengue. Therefore, confirmation is necessary through specific diagnosis [5]. In order to establish infection in VL patients, the gold standard of diagnosis is to envisage parasites through the microscopic examination of tissue aspirates. However, the process takes time and considerable levels of skill are needed to conduct the test [6]. In terms of diagnosis under laboratory conditions, immunoassays such as ELISA and direct agglutination test (DAT) have been popular methods for decades. Nevertheless, these assays have limitations, such as being time-consuming, variable in performance and not suitable for a single sample test [7]. Molecular tests are good in terms of their specificity but are generally undertaken in a good laboratory with access to expensive equipment [8]. Besides, the turnaround for the results of these laboratory-based diagnostics may take additional time due to transporting samples from resource-constrained collection centres to the test laboratories.

The advantage of point of care tests (POCTs) has been acknowledged consensually for many clinical conditions, especially for infectious disease diagnosis [9]. The World Health Organization has recommended criteria to identify appropriate POCTs, which together are referred to as ASSURED (affordable, sensitive, specific, user-friendly, rapid and robust, equipment-free and deliverable to end-users) [10]. Lateral flow assay (LFA)-based strip tests are one of the most commonly used POCTs available commercially for several diseases including VL. A typical LFA determines antigens or antibodies in the biological samples qualitatively using colloidal-gold particles. The routinely used LFA in VL diagnosis detects antibodies in the patients’ sera against the leishmanial antigen rK39, a kinesin-related protein isolated from *L. chagasi* [11]. The diagnostic performance of rK39 LFA is appropriate in the context of the highly endemic Indian Subcontinent, which has the largest number of VL cases, but it is not satisfactory for Latin America and East African regions, especially in terms of its sensitivity [12,13]. The inconsistent performance of rK39 LFA in all VL regions has led to the development of other classes of antigens which were isolated from local *Leishmania* isolates such as rKE16, rK28 and rKLO8. The performance of these antigens in the LFA format varies in different regions for the serological diagnosis of VL [14].

In the past few years, extensive diagnostic studies have been carried out considering urine as a source of biological sample in many diseases [15]. Since a number of leishmanial antigens and human antibodies specific to these antigens are present in the infected urine samples a plethora of urine-based assays have been developed for VL diagnosis in recent times. Urine-based tests such as DAT, PCR, ELISA, and LFA have been reported with varying degrees of performance as compared to the same assay with the serum samples [16,17,18]. Diagnosis of VL through urine-based LFA utilized either newer antigens or the antigens which were used earlier for serum-based diagnosis. The results obtained, however, have discrepancies in sensitivity and specificity in different VL endemic areas, therefore, urine-based LFA has not yet established as better in performance than serum-based LFA.

In the present study, we developed *Leishmania* promastigote membrane antigens (LAg)-based LFA to detect antibodies in samples qualitatively. As the diagnostic potential of LAg has been apparent in ELISA and dipstick tests in our previous studies, here we developed the LFA format to improve its diagnostic aptitude and assessed its performance with serum and urine of Indian VL patients and with Brazilian sera [13,19,20].

## 2. Materials and Methods

### 2.1. Study Design

This study was designed to develop a rapid diagnostic test using *Leishmania* membrane antigens and to assess its performance with human serum and urine samples of Indian VL patients along with Brazilian sera. A total of 174 serum and 96 urine samples were collected for this study. In total 114 serum and 41 urine samples were collected from parasitologically proven VL cases from the School of Tropical Medicine, Kolkata, India and Rajendra Memorial Research Institute of Medical Sciences, Patna, India. Additionally, 8 serum and 12 urine samples from healthy individuals were obtained from relatives of the patients as endemic controls; 20 serum and 18 urine samples were also collected from diseases other than VL, including four each of malaria, tuberculosis and typhoid, three of dengue, two of gastroenteritis and influenza-like illness along with two serum samples of filariasis and systemic lupus erythematosus. In total, 32 sera and 25 urine samples from healthy persons were acquired from CSIR-Indian Institute of Chemical Biology Kolkata, India. Parasitologically confirmed sera from 35 Brazilian VL patients and 19 healthy controls were provided by the Universidade Federal do Piaui, Teresina, Brazil.

### 2.2. Antigen Preparation

For the preparation of leishmanial membrane antigens (LAg) from the *Leishmania donovani* strain (ATCC^®^ PRA-413™), promastigotes were sedimented and suspended in chilled 5 mM Tris-HCl buffer (pH 7.6). The surface of the parasites became leaky after vortexing six times for 2 min, with each iteration resulting in the release of cytoplasmic matrix. Following this, the ghost membrane with bound proteins was obtained as pellet through the centrifugation (Eppendorf AG, Hamburg, Germany) of the suspension at 2310× *g* for 10 min at 4 °C. Subsequently, the pellet was dissolved in chilled Tris-HCl and sonicated (30 s pulse, 1 min interval, 6 cycles at 4 °C) (Misonix, Farmingdale, NY, USA) to release the proteins from the membrane. Proteins were then collected in the supernatant after centrifugation at 5190× *g* for 30 min at 4 °C. Lowry’s method was used to estimate the final concentration of the protein, and the specific pattern of the predominant proteins in LAg was confirmed through SDS-PAGE (Bio-Rad, Hercules, CA, USA).

### 2.3. Preparation and Assembly of the LFA Device

The LFA was prepared using four kinds of membranes: a nitrocellulose membrane, conjugate pad, sample pad and absorbent pad. All membranes used in the LFA were purchased from mdi Membrane Technologies, Ambala Cantt, India. Antigen, LAg (1 mg/mL for serum and 1.5 mg/mL for urine) and rabbit anti-mouse IgG (1 mg/mL) (Southern Biotech, Birmingham, AL, USA) were coated in parallel on the test and control line of the nitrocellulose membrane (5 µm, 26 × 2.4 cm), respectively, using Flowline F100 dispenser (Precore Solutions, Kochi, India) at a dispense rate of 5 µL/cm. The membrane was then dried at 37 °C for 30 min and stored at room temperature with desiccant. Colloidal gold conjugated protein G (0.5 µg/mL) (Ubio Biotechnology Systems, Cochin, India) was diluted (1:2) with 10% sucrose (HiMedia Laboratories, Mumbai, India) solution in 0.02 M PBS, and 20 µL/cm solution was spread onto the conjugate pad (26 × 0.8 cm) followed by drying with a hair dryer for 15 min. The coated conjugate pad was pasted towards the test line side with a 2 mm overlap with the NC membrane. The sample pad (26 × 2 cm) was used without treatment and pasted with a 2 mm overlap with the conjugate pad. The absorbent pad (26 × 2.5 cm) was used towards the control line at the end of the NC membrane. Prepared LFA was cut into 3 mm strips, assembled in a plastic cassette (mdi Membrane Technologies, Ambala Cantt, India) and stored at room temperature under desiccation before testing.

### 2.4. Principle of the Test

In the assay, 10 µL of serum or 20 µL urine sample was applied to the sample pad region of the cassette followed by two drops of chase buffer (20 mM TBS with 0.05% Tween 20 (Bio-Rad, Hercules, CA, USA) at pH 7.4 for serum and 10 mM Tris (HiMedia Laboratories, Mumbai, India) with 0.05% Tween 20 at pH 8.8 for urine). Antibodies present in the sample flowed towards the conjugate pad and bound with the protein G-colloidal gold complex. The entire complex with antibodies migrated towards the NC membrane. If the sample carried LAg-specific antibodies, they bound at the test line and resulted in a dark red color due to the presence of gold; Otherwise, the entire complex moved forward and bound with the control line in all cases. Excess reagents and buffer were absorbed by the absorbent pad. The reading could be interpreted after 2 min of the assay, where two colored bands at the test and control line determined a kala-azar-positive sample. One colored band at the control line confirmed only that sample was negative for kala-azar.

### 2.5. Result Interpretation

After each assay, the result was seen and recorded by at least two observers who were not involved in the assay. The final results with serum and urine samples were compiled to establish the sensitivity and specificity of the LFA.

### 2.6. Quantitative Band Intensity Analysis

The intensity of the color generated after the LFA reaction was captured using ImageJ software 1.x (National Institutes of Health (NIH), Bethesda, MD, USA). For a single set of experiment a fixed area (count) was selected for all the test and control lines and the mean of intensity was acquired. As a control level of intensity, an area anywhere on the membrane between the test and control line was selected, and the relative intensity of the test and control line was calculated in reference to the control intensity. A histogram plot was obtained to depict the intensity of the band on a scale from 0 to 255. The relative mean intensities of the test line were used to create a dot plot in GraphPad Prism software, version 5.0 (San Diego, CA, USA) 

## 3. Results

### 3.1. Optimization of Serum-Based Lateral Flow Assay

The LFA test was designed to be a field-adaptable tool for the diagnosis of VL using patients’ serum samples. The diagram of a typical LFA test is depicted in Figure 1A. The rule of thumb for a standard assay presents a clear test and control line with VL-positive samples without cross reactivity with VL-negative samples. However, within the visible test line, the intensity of the band color was further compared with ImageJ software to investigate the optimal conditions of the test quantitatively. The optimal concentration of antigen, LAg, on the test line was shown to present the strongest signal at 1 mg/mL (Figure 1B). The serum sample was optimized without any dilution so that the additional step of dilution could be avoided. The lowest volume of sample needed to obtain test line with a good intensity and a clear control band was found to be 10 µL of serum for each test (Figure 1C). The composition and pH of the chase buffer in the LFA are important to achieve adequate antigen–antibody binding without false positives. For this purpose, several buffers were tested, and we finally chose 20 mM Tris buffer saline with 0.05% Tween 20, which showed a test line with good intensity and an acceptable flow rate at pH 7.4 (Figure 1D,E).

### 3.2. Optimization of Urine-Based Lateral Flow Assay

The optimization of the urine-based assay was difficult in comparison to that of serum. Since the pH of urine is slightly acidic with abundant nitrogenous by-products, the ideal condition for the antigen–antibody reaction on the membrane was optimized accordingly. Unlike dry gold conjugate, at first, the test was conducted under wet conditions using protein G-gold and urine samples together (Figure 2A). The set condition was further developed into the LFA format. The smallest concentration of antigen, LAg, was found to be 1.5 mg/mL for the assay (Figure 2B). Different volumes of urine were tested to evaluate the effective detection level, and 20 µL of sample per test was found to be sufficient for the optimal test line intensity (Figure 2C). Chase buffer plays a vital role in the LFA. We investigated different buffer solutions for urine-based LFA and finally selected 10 mM Tris buffer with 0.05% Tween 20 at pH 8.8 for the best results (Figure 2D,E).

### 3.3. Performance of LFA with Human Sera

In order to show the diagnostic performance of optimized LFA, serum samples were employed from Indian and Brazilian VL patients along with non-VL controls. Of the 114 *L. donovani*-infected patients from India, 110 tested positive, suggesting 96.49% sensitivity. The specificity of the test was validated with 60 non-VL samples including endemic healthy controls, non-endemic healthy controls and diseases other than VL. Two samples from endemic healthy controls and one sample from malaria tested as false positives in the control group. Thus, the overall specificity of the test showed 95% accuracy. The sensitivity of the LFA was also demonstrated by testing 35 Brazilian VL sera. In total, 31 serum samples were found to be positive in the LFA, demonstrating 88.57% sensitivity. The LFA was specific up to 94.73% for *Leishmania infantum* infection and cross-reacted with only one healthy serum sample out of 19 (Table 1).

### 3.4. Performance of LFA with Human Urine Samples

To investigate LFA for non-invasive diagnosis, we investigated Indian VL urine samples. Of 41 urine samples that were tested by LFA, 39 samples were seen to be positive. Therefore, the sensitivity of the assay was calculated as 95.12%. In terms of specificity, a total of 55 non-VL urine samples were examined, including 25 healthy controls, 12 non-endemic healthy controls and 18 samples from other symptomatic diseases. Two malaria samples showed cross reactivity with the LFA test, thus showing 96.36% overall specificity, with 100% specificities with endemic and non-endemic healthy controls (Table 1).

### 3.5. Quantitative Detection of Band Intensity

The test line color intensities of all LFAs conducted with VL-positive and negative samples were subjected to image analysis and the relative intensity for each test was scored quantitatively. The representative results of LFAs with sera of four confirmed Indian VL cases and four healthy controls with their histogram plots are shown in Figure 3A,B. Histogram plots were also plotted with a positive and a negative urine sample as well as with a Brazilian VL infected and a healthy serum (Figure 3C,D). The color obtained from the test line showed prominent intensities with VL-positive samples as compared to the healthy samples. The control line intensity in a similar set of assays was found to be comparable in image analysis. The relative mean intensities of the test lines of 24 positive and 18 negative healthy Indian sera (Figure 4A) and 20 VL and 20 healthy urine samples were plotted in the dot plot graph (Figure 4B). At the acquired cut-off, the mean band intensities of positive test lines were significantly higher than the negative test lines in both serum and urine LFAs.

### 3.6. Stability and Reproducibility

The shelf life of the LFA test was determined by the accelerated aging experiment. The LFA strips were tested and the band intensity was observed initially. Subsequently, strips were kept at 37 °C in desiccant and tested on alternate days for one week. The LFA strips were also tested in real time after being stored at room temperature and 4 °C for one month. In each batch, the storage conditions for the LFA were found to be stable with the same reactivity and specificity. However, a slight decrease in intensity was observed in the test line after one week at 37 °C. Using dehumidifier conditions during strip making and storage can further increase the shelf life, which we did not verify further. For the intra-assay reproducibility experiment, the LFA test was scrutinized in triplicate, and test line intensities were compared. Samples were tested in three different sets of LFAs consecutively for inter-assay reproducibility and were found to have acceptable intensity.

## 4. Discussion

Visceral leishmaniasis (VL) manifestation due to *Leishmania* is still an important health issue in developing countries, mainly affecting poor people. In recent years, several studies on VL diagnosis have been carried out using immunological and molecular techniques as point-of-care tests. Since the disease mostly affects developing countries where health facilities are constrained for all areas, POCs can have the advantages of onsite diagnosis and result interpretation. In this study, we developed a colloidal-gold-based lateral flow test using leishmanial antigen, LAg, and evaluated its diagnostic ability with Indian serum and urine samples and with Brazilian sera.

In the last decade, rK39 antigen-based immunochromatographic tests have shown remarkable results in VL diagnosis, especially in the Indian Subcontinent. However, concerns regarding their use in other endemic areas have been raised due to their low sensitivity. To deal with the limitations of rK39 antigen-based tests, researchers in the past few years have managed to develop several other recombinant antigens. These antigens were either isolated from native *Leishmania* species or a chimera of selected antigens [21,22,23]. However, the variable sensitivity of recombinant antigens has been attributed to the diversity of the protein in different *Leishmania* species as well as the occurrence of different antibody titers in VL patients across different geographical regions [24]. Additionally, unlike purified native antigens, recombinant antigens lack the post-translational process responsible for necessary protein modification with carbohydrate and lipid moieties. This may be the reason why the crude antigens used in DAT show higher specificity than rK39 recombinant antigen [25]. The results of our previous studies have validated the diagnostic potential of *L. donovani* purified antigen, LAg, in ELISA, immunoblot and dipstick assays using patients’ sera and urine. The sensitivity and specificity obtained with LAg have shown excellent performance in detecting antibodies in *Leishmania*-infected samples. Recently, a multicentric serological study of an LAg-based dipstick in six countries—India, Nepal, Sri Lanka, Brazil, Ethiopia and Spain—demonstrated an overall sensitivity of 97.10% with 93.44% specificity [19]. Moreover, the LAg-based dipstick appeared to be 100% sensitive and specific with Indian urine samples [13]. The current work represents the next step of our earlier study aiming at the use of thispurified antigen, LAg, to develop point-of-care tests for VL diagnosis.

In the present study, we determined the optimal conditions for LFA using sera and urine from VL patients. Although the results of the LFA can be interpreted visually, minor differences in the band intensities are not possible to distinguish by the naked eye. The present study describes an approach to optimize the LFA test by quantifying the band intensities by ImageJ software. Comparing the values of band intensities can help to choose an optimal condition.

The developed LFA test was validated by comparing the results of *Leishmania*-infected Indian sera with healthy and other diseased sera. The test was found to have an overall sensitivity and specificity of 96.49% and 95%, respectively. Two endemic control sera and one malaria serum showed cross-reactivity with LFA. However, the test line bands for endemic controls were very faint. Apart from a visual observation of the bands, the LFA test was subjected to image analysis. The dot plot was drawn with the test line intensity reading of VL-positive and negative sera and a cut-off was set. This could be helpful to come to a conclusion in the condition of a result that is visually unclear to an observer.

Despite the low sensitivity in Brazil, rK39-based tests are routinely used to investigate suspected VL cases. In a recent study in Brazil, a rK39 test showed 81% sensitivity and 96% specificity in comparison to DAT, which showed a sensitivity and specificity of 87% and 100%, respectively [26]. Reports on rK39 RDT with different biological samples demonstrated 89.8% sensitivity and 96.3% specificity with the Brazilian sera [27]. In this study with LAg-based LFA, we showed 88.57% sensitivity and 94.73% specificity using Brazilian sera. In light of the comparable results obtained by our LFA, it would be useful to diagnose VL caused by *L. infantum*. However, further studies utilizing a greater number of samples are warranted.

The invasiveness of blood sampling represents a barrier to diagnosis as patients are often hesitant to donate blood. Therefore, non-invasive samples such as urine and oral fluid have been evaluated in diagnosis due to the ease of their collection and storage in contrast to serum. rK39 rapid tests for diagnosing Indian VL through urine samples showed 96.1% to 100% sensitivity in total [28]. However, the rationale behind using urine samples in VL diagnosis is not conclusive to date. In this work, we optimized the LFA for the urine-based diagnosis of VL. Our study showed 95.12% sensitivity and 96.36% specificity for a LAg-based LFA with urine samples.

Despite the good performance of the LAg-based LFA, we recognize certain limitations of the present study. A more general limitation of the purified antigens is the batch-to-batch variation in its production, although under the standardized protocol, we isolated LAg with negligible variation. Additionally, the cross-reactivity of the test was not evaluated in Brazil against other disease samples. Therefore, in the next phase of the study, more categorized patient groups will be included within endemic areas, and the test performance will be compared against other diagnostic tests and recombinant antigens to enhance the reliability of the assay.

## 5. Conclusions

In conclusion, a rapid immunochromatographic test was developed based on the leishmanial antigen LAg, and its diagnostic potential was determined. The LAg-based LFA showed promising diagnostic performance in India when utilizing human sera as well as urine. The test also showed its ability to detect Brazilian VL serologically. This LFA test has benefitted from the fact that it is a ready-to-use device that was optimized and ready to validate in real clinical settings. The study also highlights the application of the test for point-of-care diagnosis without the need for sample processing, specialized laboratory equipment and a trained person. Additionally, as an alternative to visual detection, we illustrated the quantification of the band-intensities for subtle assay optimization. Considering the advantage of the current findings, this indigenous test is eligible for further study with large patient cohorts in field settings. In the future, the test can contribute as an alternative to current diagnosis and eventually participate in VL management and control programs.

## Figures and Tables

**Figure 1 microorganisms-09-01369-f001:**
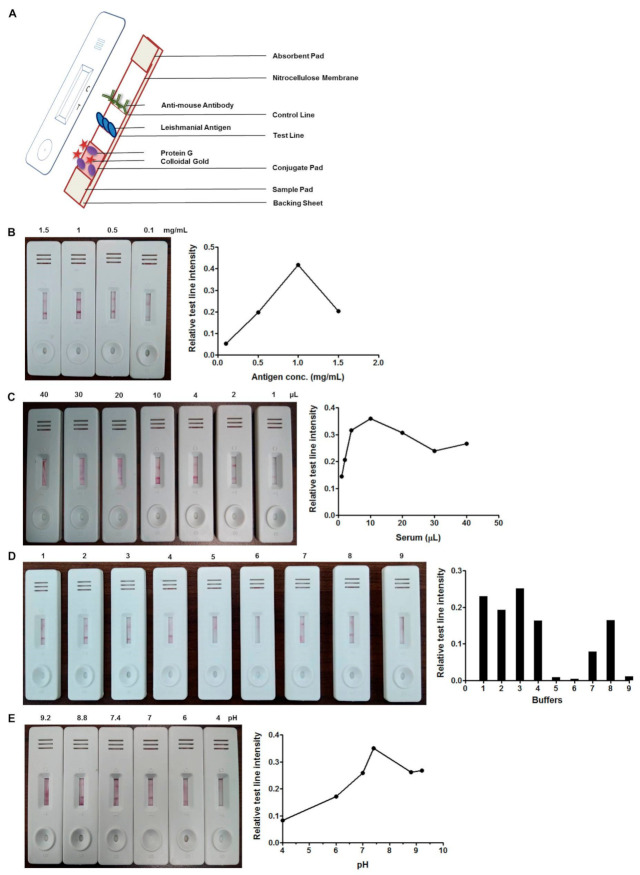
A representative diagram of a typical lateral flow assay (LFA) (**A**). Optimization of LFA for serum-based diagnosis of visceral leishmaniasis (VL) with 0.1, 0.5, 1 and 1.5 mg/mL of leishmanial antigen, LAg (**B**). Standardization of the volume of undiluted sera with 1, 2, 4, 10, 20, 30 and 40 µL of serum per test (**C**). Selection of the optimal chase buffer for the assay, 1, 10 mM Tris; 2, 10 mM Tris + 0.05% Tween 20; 3, 20 mM Tris buffer saline + 0.05% Tween 20 (TBST); 4, buffer provided with rK39 test; 5, 20 mM borate buffer; 6, 20 mM phosphate buffer saline (PBS); 7, 20 mM Hepes; 8, 5% BSA + 9% NaCl + 0.05% Tween 20; 9, distilled water (**D**). LFA using TBST chase buffer at different pH levels: 9.2, 8.8, 7.4, 7, 6 and 4 (**E**). The graphs plotted on the right denote the relative means of the test line intensities obtained by ImageJ 1x (National Institutes of Health (NIH), Bethesda, MD, USA).

**Figure 2 microorganisms-09-01369-f002:**
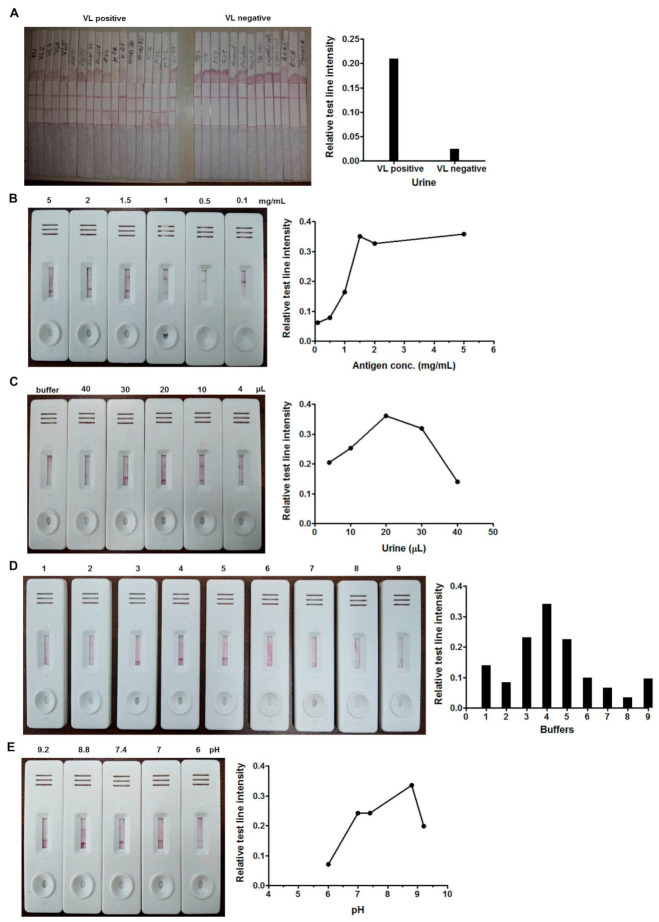
Lateral flow assay (LFA) under wet condition with VL-positive and negative samples (**A**). Optimization of LFA for urine-based diagnosis of VL with 0.1, 0.5, 1, 1.5, 2 and 5 mg/mL of leishmanial antigen, LAg (**B**). Standardization of the volume of 4, 10, 20, 30 and 40 µL of undiluted urine sample per test including buffer as a control (**C**). Selection of the optimal chase buffer for the assay, 1, 10 mM Tris; 2, 20 mM borate buffer; 3, buffer provided with rK39 test; 4, 10 mM Tris + 0.05% Tween 20; 5, 20 mM Tris buffer saline + 0.05% Tween 20 (TBST); 6, 20 mM phosphate buffer saline (PBS); 7, 20 mM Hepes; 8, distilled water; 9, 5% BSA + 9% NaCl + 0.05% Tween 20 (**D**). LFA using chase buffer 10 mM Tris with Tween 20 at different pH levels: 9.2, 8.8, 7.4, 7 and 6 (**E**). The graphs plotted on the right denote the relative means of the test line intensities obtained by ImageJ.

**Figure 3 microorganisms-09-01369-f003:**
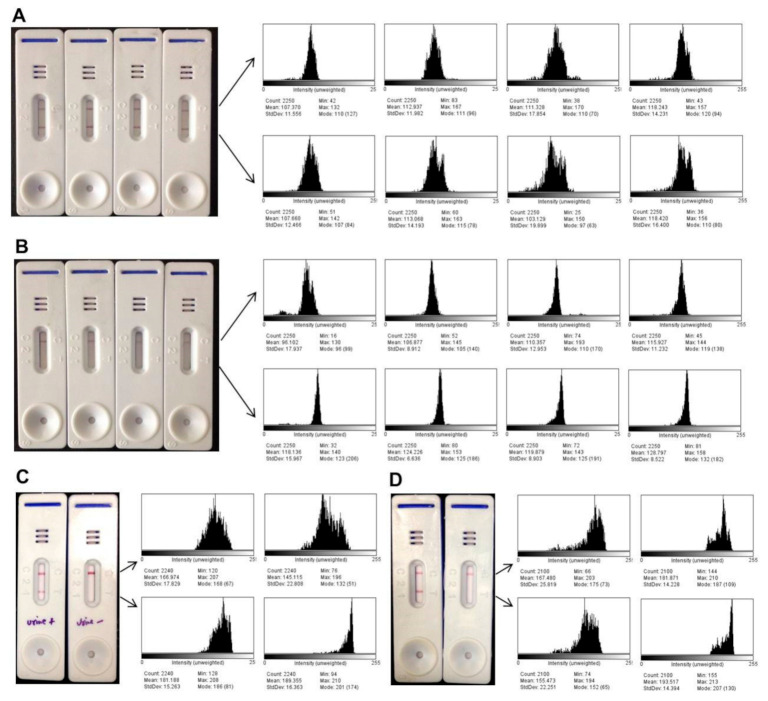
Representative results of lateral flow assays (LFAs) with four visceral leishmaniasis (VL) positive (**A**) and four VL negative control sera (**B**), a VL-positive and a negative urine test (**C**) from India, and LFAs with a Brazilian VL positive and negative serum samples (**D**). The histogram plots on the right denote the means of the test and control lines intensities obtained by ImageJ.

**Figure 4 microorganisms-09-01369-f004:**
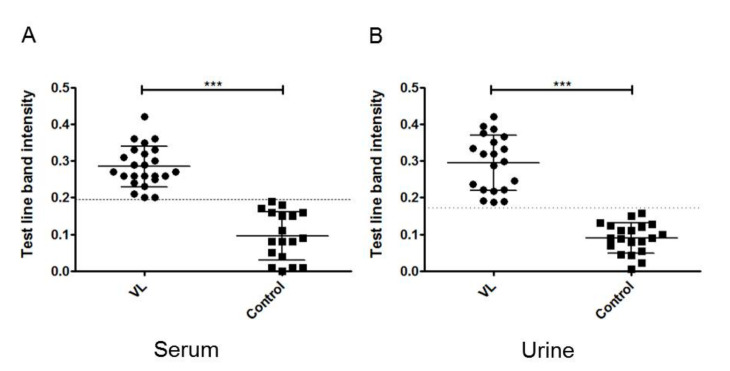
Dot plot graph obtained by the analysis of test line intensities of 24 visceral leishmaniasis (VL)-positive and 18 healthy Indian sera (**A**) and 20 VL-positive and 20 healthy urine samples (**B**) in lateral flow assay. The cut-offs (dotted lines) were obtained by GraphPad prism and the receptor operative characteristic (ROC) curve, at which the maximum sensitivity and specificity were achieved. Two-tailed Mann–Whitney U tests were performed to compare the intensities of VL-positive and healthy control samples, where *p* values < 0.05 were statistically significant (*** represents *p* values < 0.0001). Each dot denotes the test line intensity obtained from a single sample and the solid horizontal lines show mean ± standard error of mean (SEM).

**Table 1 microorganisms-09-01369-t001:** Sensitivity and specificity of lateral flow assay tests.

Sample	Country	Sensitivity to Active VL Patients in % (n/N)	Specificity to Non-Endemic Healthy Controls in % (n’/N)	Specificity to Endemic Healthy Controls in % (n’/N)	Specificity to Other Diseases in % (n’/N)	Total Specificity
Serum	India	96.49 (110/114)	100 (32/32)	75 (6/8)	95 (19/20)	95 (57/60)
	Brazil	88.57 (31/35)	94.73 (18/19)	-	-	94.73 (18/19)
Urine	India	95.12 (39/41)	100 (25/25)	100 (12/12)	88.88 (16/18)	96.36 (53/55)

Abbreviations: VL, visceral leishmaniasis; n, number of positive samples in each group; N, total number of samples tested in each group; n’, number of negative samples in each group.

## Data Availability

All data generated or analyzed during this study are included in this article.

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
