# Peer review of "Development and Clinical Evaluation of Serum and Urine-Based Lateral Flow Tests for Diagnosis of Human Visceral Leishmaniasis"

_microorganisms, 2021, doi:10.3390/microorganisms9071369_

Round 1

Reviewer 1 Report

In this work, Ejazi and collaborators describe the construction and development of an LFA device for the diagnosis of human VL. The source of antigen is a protein extract obtained from Leishmania donovani, which has been previously validated in other studies carried out by the authors using other immunological techniques. The diagnostic potential of the device is analyzed using serum and urine samples obtained from patients in two different continents. As the diagnostic potential of the protein preparation of the parasite has been assessed in previous studies, the scientific novelty of the work is not very high (this is the main drawback). The greatest strength of the work is the importance of achieving a reliable and field-employable diagnostic assay for human beings. The work is well written and referenced, although the authors should take into consideration its edition to clarify the following aspects.

Abstract.

The text of the abstract must be changed when talking about the extract used. As it is written, it seems that a single antigen is used instead of a protein preparation. It must be clearly indicated that a previously analyzed protein fraction extract is being used.

Considered editing the next sentence 22-24: The intensity of the bands depicting antigen-antibody complex was optimized quantitatively under different experimental conditions and “QUANTITATIVELY ANALYZED” by the ImageJ software.

Introduction.

Please, include the corresponding references in lines 82-84: Since in our previous studies the diagnostic potential of LAg has been apparent in ELISA and dipstick test (REFERENCES),…

Other sections.

More detailed information on the patients employed needs to be included. What pathologies do the patients in the other diseases group have? What number of sera belong to this group and what number to the healthy group?

Please, include the volume data of Lag applied in each blot line if you wish to keep the concentration data in both sections (including the captions of figure 1 and 2). It would be enough to include it in the Mat. and Meth. section. Include this information also for the control IgG line. Please note that in the membrane the protein is not in solution, so the concentration data generates confusion. Alternatively, these data can be included with units of mass. Please, also indicate the concentration of the colloidal gold conjugated protein G and the volume employed for spreading the the conjugate pad.

For the statistical analyses, include in Mat. and Meth. that the GraphPad software is used. In Figure 4 indicate with analysis was employed to define the cut off (ROC Curve? Other?) and to compare VL and control relative intensity. What data were employed for the cut-off calculation, healthy? Other diseases? Both? Please discuss this important aspect in the text (results or discussion). In addition, include in the figure 4 legend what the bars show. Mean with SEM? SD? Other?

Figure 2, panel A. It should be more informative to include a dot blot graph (like this shown in Figure 4).

Section 3.6. These data should be reported using ImageJ data and dot plots (for example as supplementary material) to avoid the use of expressions such as: “slight decrease in intensity” (244) or “found to have acceptable intensity” (249).

Minor.

 Line 96. Consider changing pellet (pelleted) by sedimented and pellet by sediment.

Include a space between value and subunit throughout the text; for example: 2 mg/ml instead 2mg/ml.

Please, use the italic fonts for species in 195, 202 and in the reference list.

Reviewer 2 Report

The manuscript from Ejazi et al describes the development and clinical evaluation of serum and urine-based lateral flow test for diagnosis of visceral leishmaniasis. Overall the manuscript is very well written and data are straightforward. The authors propose the development of a lateral flow test based on crude antigen from promastigotes as alternative for recombinant antigens. Some issues were raised during the review, as indicated below:

  1. Introduction. The authors claim that results with kinesin-derived antigens and other well established antigens for the serodiagnosis of visceral leishmaniasis are inconsistent. While variation indeed occurs in endemic areas it might quite be associated with time of infection, progression of disease, parasitic burden as other factors. Please clarify what would be the inconsistencies (or why it is not satisfactory) as indicated by the authors.
  2. The use of extract product from membrane of promastigotes might offer a variety of antigens and, of course, a better performance (mainly in sensitivity) would be highly expected. Moreover, the production of these antigens seems feasible and possibly cheaper than recombinant antigens. The question point, though, is that use of promastigote-derived antigens might render cross-reactivity (with other infections) and lack of reproducibility. 
    1. The authors did not provide any information on cross reactivity with other diseases. The calculation of cross reactivity should be made by using samples from patients with other etiologies (mainly those that might occur in the endemic areas). Cross reactivity was just calculated by using samples from negative individuals and, as it is, it is just a preliminary result.
    2. While authors provided inter assay reproducibility with clear results, the main concern when crude antigens are used is the reproducibility between different batches of antigen production. This should be demonstrated to justify the use of crude antigen in the test.
  3. Ethical approval. Please provide the protocol number of approval from both IRBs (India and Brazil) and where it was approved.
  4. Urine LFA test. What was the purpose of the use of urines for this test? It was to detect antibodies presented in the urine? This condition should be expected only at the severe disease.
  5. The authors indicated that "Unlike dry gold conjugate, at first, the test was conducted under wet condition using protein G-gold and urine samples together (Figure 2A)" (Lines 173-174). The use of wet condition provides a better signal but it is definitely the final condition of testing. It is not clear whether the presented results are from wet condition or dried condition. As the authors know, the dry condition may decrease the signal and, considering it is the final condition of the test, the real results should be obtained under this condition.
  6. Figure 4. As the manuscript is not clear about the statistical analysis, please provide in the legend the statistical test used and if Gaussian distribution was tested before.
  7. The discussion section is a mere repetition of the results. The authors should consider to discuss on how their results might contribute to the better diagnosis of visceral leishmaniasis. They should emphasize the strength of the work (sensitivity and detection in different endemic areas of New and Old World) but also the limitations. Among them, the possible lack of reproducibility (between different batches), the cross reactivity, the comparative results with other recombinant antigens, etc.

Reviewer 3 Report

The problem of leishmaniasis is current and of great importance, the countries affected by this pathology are numerous, there has been an increase in the Mediterranean countries with the appearance of subclinicians, especially in endemic areas such as Brasil and india, both in terms of sensitivity and speed, but the use of urine is equally useful. The experimental design is correct and well conducted. The results and conclusions are clear and in line with the experimental design.

Author Response

Point-by-point Response to the Reviewer’s Comment

# Reviewer 2

The problem of leishmaniasis is current and of great importance, the countries affected by this pathology are numerous, there has been an increase in the Mediterranean countries with the appearance of subclinicians, especially in endemic areas such as Brasil and india, both in terms of sensitivity and speed, but the use of urine is equally useful. The experimental design is correct and well conducted. The results and conclusions are clear and in line with the experimental design.

Response: Authors thank the reviewer for his/her appreciation. Reviewer is correct. The disease is increasing in the newer areas of the world such as in South Europe and in the Mediterranean countries. Moreover, urine is a non-invasive sample thus can be easily collected and stored. Thanks for praising the current work.